# Genetic Determinants of Atherogenic Indexes

**DOI:** 10.3390/genes14061214

**Published:** 2023-06-01

**Authors:** Tomas Texis, Susana Rivera-Mancía, Eloisa Colín-Ramírez, Raul Cartas-Rosado, David Koepsell, Kenneth Rubio-Carrasco, Mauricio Rodríguez-Dorantes, Vanessa Gonzalez-Covarrubias

**Affiliations:** 1National Institute of Genomic Medicine (INMEGEN), Mexico City 14610, Mexico; tomastexisvalencia@gmail.com (T.T.); krubio_carrasco@hotmail.com (K.R.-C.); mrodriguez@inmegen.gob.mx (M.R.-D.); 2National Institute of Cardiology Ignacio Chavez, Mexico City 14080, Mexico; susana.rivera.mancia@gmail.com (S.R.-M.); rcartas@gmail.com (R.C.-R.); 3School of Sports Sciences, Anahuac University of North Mexico, Huixquilucan 52786, Mexico; ecolin@anahuac.mx; 4Conduct Research Committee, Texas A&M University, College Station, TX 77843, USA; drkoepsell@tamu.edu; 5Faculty of Chemistry UNAM, Mexico City 04510, Mexico

**Keywords:** dyslipidemia, atherogenic indexes, genetics

## Abstract

Atherogenesis and dyslipidemia increase the risk of cardiovascular disease, which is the leading cause of death in developed countries. While blood lipid levels have been studied as disease predictors, their accuracy in predicting cardiovascular risk is limited due to their high interindividual and interpopulation variability. The lipid ratios, atherogenic index of plasma (AIP = log TG/HDL-C) and the Castelli risk index 2 (CI2 = LDL-C/HDL-C), have been proposed as better predictors of cardiovascular risk, but the genetic variability associated with these ratios has not been investigated. This study aimed to identify genetic associations with these indexes. The study population (n = 426) included males (40%) and females (60%) aged 18–52 years (mean 39 years); the Infinium GSA array was used for genotyping. Regression models were developed using R and PLINK. AIP was associated with variation on *APOC3*, *KCND3*, *CYBA*, *CCDC141/TTN*, and *ARRB1* (*p*-value < 2.1 × 10^−6^). The three former were previously associated with blood lipids, while CI2 was associated with variants on *DIPK2B*, *LIPC*, and 10q21.3 rs11251177 (*p*-value 1.1 × 10^−7^). The latter was previously linked to coronary atherosclerosis and hypertension. *KCND3* rs6703437 was associated with both indexes. This study is the first to characterize the potential link between genetic variation and atherogenic indexes, AIP, and CI2, highlighting the relationship between genetic variation and dyslipidemia predictors. These results also contribute to consolidating the genetics of blood lipid and lipid indexes.

## 1. Introduction

Atherogenesis and dyslipidemia are key risk factors for coronary artery disease (CAD), the leading cause of mortality in the world [1]. Atherosclerosis is an inflammatory process that encompasses the formation of plaque in the artery walls, contributing to cardiovascular disease (CVD), hypertension, stroke, and coronary artery disease [2]. Atherosclerosis is a complex and multifactorial disease, underlined by genetics, the environment, low-density lipoprotein cholesterol (LDL-C), total triglycerides (TG), and high-density lipoprotein cholesterol (HDL-C), oxidized LDL, and the comorbidities diabetes, dyslipidemia, and hypercholesterolemia [3]. Increasing age is highly correlated to atherosclerosis, and current epidemiological studies suggest that early detection can lead to prevention and the deployment of interventions for treatment and control prior to cardiovascular disease onset. However, the early characterization of the atherosclerotic phenotype at the molecular level is incomplete and ought to be fully elucidated [4]. Dyslipidemia, one of the main underlying factors in atherogenesis, is highly prevalent worldwide; for instance, it reaches 53% in the United States, 49% in China [5], and 80% in Turkey [6]. In Mexico City, the CARMELA study determined a prevalence of 50.5% for hypercholesterolemia and 32.5% for hypertriglyceridemia, and the hypotheses are that nearly every two out of three city dwellers might have some type of dyslipidemia [7,8].

The early identification of atherosclerosis and dyslipidemia is urgently needed to pinpoint targeting therapies and preventive interventions. Lipid levels and lipid indexes have been of some help because of their association with cardiovascular risk. For example, LDL-C, TG, and HDL-C are strongly correlated to coronary heart disease (CHD), although their underlying genetics have not been fully characterized [9]. A genetic profile that may aid in the preemptive identification of dyslipidemia, CVD, or an individual’s predisposition could propel earlier diagnosis and preventive interventions.

To date, the association between genetics and blood lipids has aided in the search for reliable markers of CVD, atherosclerosis, and dyslipidemia since, in part, these have a heritable basis and the literature has evidenced their association with an extensive collection of genetic loci [10,11]. For example, single nucleotide variants (SNVs) of *APOE*, *CETP*, *LPL*, *PCSK9*, and *GCKR* have been significantly associated with lipid levels and dyslipidemia in different populations [10,11]. Despite the high prevalence of CVD and dyslipidemia in Mexico, only a few studies have investigated its relationship with genetic variation. One report found SNVs on *APOA5*, *GCKR*, *LPL*, and *NPC1* associated with hypertriglyceridemia [12], while polymorphisms on *ABCA1*, *CETP*, *LIPC*, and *LOC55908* have been associated with hypoalphalipoproteinemia. Many of these variants are shared by different populations, but some seem to be unique to certain geographical ancestries [10,13].

Recent studies have highlighted the accuracy and relevance of lipid ratios/indexes to better assess dyslipidemia and cardiovascular risk. The atherogenic index of plasma (AIP = log TG/HDL-C) can accurately predict hypertension, metabolic syndrome, and ischemic stroke, even when HDL-C and TG levels seem normal or when isolated values of TG or HDLC-C cannot assess this risk [14,15]. The direct measurement of HDL-C and LDL-C has shown bias in assessing cardiovascular health [16], but their ratio, i.e., the Castelli Index I2 (CI2 = LDL-C/HDL-C), although less cited, has been confirmed as a better predictor of cardiovascular risk [9,16,17]. Hence, there is an apparent, but not as frequently acknowledged, value of the AIP and CI2 indexes to identify cardiovascular risk. It is possible that genetic variants associated with AIP and C2 ratios could hint at biochemical paths linked to the development of CVD risk and dyslipidemia. Nevertheless, little is known about the direct relationship between these indexes and genetic variation. Therefore, we investigated the potential association between lipid indexes, AIP and CI2, and genetic variants in Mexican adults free of cardiovascular disease. The identification of molecular markers, such as genetic variants associated with these indexes, may provide tools for detecting atherogenesis in a preemptive manner.

## 2. Materials and Methods

### 2.1. Study Population

Study participants were volunteers recruited between 2014 and 2016 for the longitudinal study “Tlalpan 2020” (n = 426) [18], and all were normotensive and had no previous diagnosis of cardiovascular disease (Table 1). The study protocol followed the principles of the Declaration of Helsinki and was approved by the Institutional Bioethics Committee of the Instituto Nacional de Cardiologia numbers 13–802 and 16–983 and at INMEGEN, CEI2017/20. A blood sample was withdrawn after an overnight 12 h fast in EDTA-Vacutainers. Anthropometric measurements included height, weight, waist, and hip circumference. Blood pressure was registered as the average of three measurements with a calibrated sphygmomanometer. Clinical determinations were performed for triglycerides (TG), HDL-C, LDL-C, uric acid, creatinine, and glucose; levels were compared to reference values to define dyslipidemia (Table 1) [17,19]. Total TG, HDL-C, and LDL-C levels were used to calculate the Castelli risk index 2, CI2 = LDL-C/HDL-C [18], and the atherogenic index of plasma, AIP = log (TG/HDL-C) [14,20,21].

### 2.2. Lipid Indexes and Genetic Analyses

DNA was extracted using the PureBlood kit (Qiagen, Valencia, CA, USA), and nucleic acid quality control and concentration were assessed in a Nanodrop (ThermoFisher, Waltham, MA, USA) and diluted to 40 ng/μL. DNA samples were stored at −70 °C until analysis. Plasma was used to assess lipid values and indexes according to laboratory tests and the equations: CI2 = LDL-C/HDL-C and AIP = log (TG/HDL-C). DNA samples were genotyped for 670 K variants using the GSA-Infinium array 24 v.10 (Illumina). After bioinformatic quality control, we excluded redundant SNVs, variants with less than a 95% genotype call rate, missing data per variant > 5%, missing data per individual > 2%, and minor allele frequency MAF < 1%; for statistical analyses, we considered 330 K variants based on PLINK algorithms [22].

Independent linear regression models were developed with AIP and CI2 as dependent variables, assuming additive effects on the allele dosage and selected covariates using PLINK and R [22,23]. Descriptive genomic analyses included call rate > 0.95, sex check by heterozygosity, and Hardy–Weinberg equilibrium considering a *p*-value < 1.0 × 10^−5^ for the latter [24]. Model covariates were selected based on a PCA analysis and their impact on the total variance for AIP or CI2. The model covariates were uric acid, weight, waist circumference, and sex.

To account for population stratification, we assessed genetic admixture using Software Admixture 1.3 [25], and the 1000 G project reference populations, Northern Europeans from Utah (CEU, Caucasians), Mexicans from Los Angeles (MXL), Yoruba in Ibadan from Nigeria (YRI), Chinese Han from Beijing (CHB), and Natives from Mexico. Genetic data from the GSA microarray were used to define 56,000 Ancestry Informative Markers (AIMs), setting an identity by descent value, IBD pi-hat Z0.5, and excluding markers in linkage disequilibrium or a physical distance < 500 kb, ensuring that ancestry informative markers (AIMs) were uniformly distributed throughout the genome.

### 2.3. Regression Models

To test for an association between AIP or CI2 as continuous variables and 330 K genetic variants, we developed linear regression models based on classical GWA strategies [26,27,28] using PLINK. These models, one for AIP and another for CI2, assumed additive effects for each SNV tested. Confounding parameters were included in the model as covariates and were selected based on their correlation with the first component of the PCA, which considered its impact on the AIP and CI2 phenotypes. Final covariates included uric acid, weight, waist circumference, and sex. We also accounted for population stratification by including the Native ancestry proportion as a covariate in the model. Models were developed using R and PLINK [22,23]. Statistical significance was considered at *p*-value = 1.0 × 10^−5^ given that our population was tested for 1.0E5 SNVs (330 K variants). Our study was defined as exploratory, so genotype–phenotype associations were not corrected for multiple testing; consequently, they will require future validation.

## 3. Results

### 3.1. Population and Lipid Characteristics

The study group consisted of 40% males (n = 170) and 60% females (n = 256), with a mean age of 39 years (17–53 year). Mean and median values were within the reference laboratory ranges for uric acid, creatinine, glucose, HDL-C, LDL-C, and TG (Table 1). Males showed higher total TG (49%), LDL-C (32%), total cholesterol (50%), and HDL-C (45%) levels compared to females (*p*-value 1.9 × 10^−2^–9.7 × 10^−4^, Table 2). Dyslipidemia was identified in 70% of males and 72% of females; deviation from reference levels is depicted in Table 2. In addition, females showed lower creatinine and uric acid levels compared to males and overall displayed a healthier lipid profile, in agreement with previous reports of lipid sex differences [16,29]. Calculations for AIP and CI2 indexes showed that men presented 26% and 18% higher. The AIP and CI2 values compared to women (*p*-value ≤ 4 × 10^−6^) reflected an expectedly higher cardiovascular risk. AIP and CI2 indexes showed a wide normal distribution, AIP: mean 0.419 ± 0.281 and range (−0.40–1.610) and CI2: mean 2.66 ± 0.841 and range (0.430–5.36) (Table 1 and Table 2).

### 3.2. Genotype–Phenotype Associations

#### 3.2.1. Atherogenic Index of Plasma and AIP Genotype–Phenotype Association 

AIP was associated with six variants on, *APOA1/APOC3*, *CYBA*, *ARRIB1*, *TTN/CCDC141*, and *KCND3* (*p*-value = 1 × 10^−6^, Table 3) and to the other 26 variants with a *p*-value ~1 × 10^−5^ (Appendix A). The visual impact of the most significant genotypes on AIP values is represented in Figure 1A. The variant allele G of *APOC3* rs5128 seemed to have a detrimental effect on AIP values, while the G allele of *ARRB1* rs11236389 was suggestive of a protective atherogenic effect showing an association with lower AIP values.

#### 3.2.2. Genotype–Phenotype Association for CI2

We found a statistically significant association between CI2 and nine variants, mostly in intergenic regions, four located on chromosome 12 in partial linkage disequilibrium (*p*-value < 1 × 10^−7^–1 × 10^−5^, Table 3, and Appendix A). The top associated variants were *APOC3* rs5128, *ARRB1* rs11236389, *LIPC/ALDH1A2* rs261342, *DIPK2B* rs4294309, and *KCND3* rs6703437. The latter was also associated with AIP, although with a lower significance (*p*-value = 1.1 × 10^6^–1.8 × 10^−5^, Table 3). Figure 1B depicts the impact of the most significant variants on CI2, where the minor allele of 10q21.3 rs11251177, and *LINC0245* rs6582413 showed an association with a 25% increment in CI2 levels (*p*-value 5.5 × 10^−5^) (Table 3).

## 4. Discussion

The identification of a quantitative relationship between genetics and CVD surrogates, such as AIP and CI2, is of health transcendence due to the prominent role of atherogenesis in cardiovascular disease, and its impact on public health. Several lipid levels and their indexes have attempted to predict cardiovascular risk and support prevention strategies. There are a couple of studies that associate lipid and lipoprotein measurements with genetic loci in different populations, but high interindividual or population variability has clouded their interpretation and potential application [30,31]. The AIP and CI2 indexes have emerged as surrogate markers of cardiovascular health, and they have been reliably correlated with cardiovascular risk, lipoprotein size [17,32], or plasma atherogenicity [30]. They have been demonstrated to be better CVD predictors compared to TG/HDL-C alone [33]. It has also been shown that the AIP index closely correlates to lipoprotein particle size and fractional esterification rate of HDL-C, which in turn is a predictor of coronary artery disease risk [31], cerebrovascular accident [34,35], the thickness of the carotid intima-media, statin response, and ischemic stroke [34]. Current reports have provided valuable molecular insights into lipid metabolic pathways and dyslipidemia, but no study has identified the relationship between the ratios AIP and CI2 with genetics [10]. Genetic variants associated with these indexes may provide tools for detecting atherogenesis in a preemptive manner and, hence, deploying preventive interventions. The clinical utility of these potential markers will rely on replication and validation studies, as well as its implementation in the clinical lab but may become tools for preemptive medicine.

Here, we report statistically significant associations between gene variants and lipid indexes AIP and CI2, some of which have been previously reported as relevant for lipid levels and CVD, including *APOC3/APOA1*, 10q.21.3 rs1125117, *KCND3*, and *VLDR*. Evidence from other fields suggests that adding genetic information to clinical interpretation may support the utility of genetic variants as predictors of atherogenesis [19] likely facilitating the development of specific laboratory tests and algorithms. Below, we discuss the relevance of our findings to the scope of novel and previous genetic associations.

### 4.1. Genetic Associations with the Atherogenic Index of Plasma, AIP = log(TG/HDLC)

The identification of variants on *APOC3* and *VLDR* associated with atherogenic indexes confirmed previous reports, since these genes are well known to impact lipid levels. *APOC3* has been repeatedly associated with dyslipidemia [36] and blood lipids. Several studies have confirmed a variety of loci, not always in linkage disequilibrium (LD), mapping on the *APOC1*, *APOC3*, and *APOA5* clusters and their relation to blood lipids [37,38,39]. Variant *APOC3* rs147210663 has been reported to be associated with dyslipidemia, cholesterol, and BMI over 40 times. It is in LD with *APOC3* rs5128 identified here, and its association with triglycerides in Pima Amerindians has been reported as a founder mutation [28]. Additionally, a recent multi-ancestry analysis on 170,000 exomes, including 16,440 individuals of “Hispanic” origin, reported that *APOC3* is a relevant gene for HDL-C and the TG/HDL-C ratio [11]. To further delve into the relevance of the *APOC3* and chromosome 11 loci, Jurado-Camacho et al. described the *APOA1/C3/A5-ZPR1-BUD13* cluster and its impact on several lipid traits, including HDL-C and TG [28]. These observations agree with our results of the intron variant *APOC3* rs5128 as significantly associated with the AIP index in Mexican adults, highlighting that the connection of *APOC3* and blood triglycerides is likely population independent [28]. Additionally, associated with AIP were intronic variants, *ARRB1* rs11236389 and *CYBA* rs12709102, the former codes for the cytosolic protein, arrestin β 1, with immune functions, but no clinical reports were found. The second one, *CYBA* rs12709102, is part of the microbicidal oxidase system of phagocytes that has also been related to CAD, the thickness of the carotid intima media, and as a direct indicator of atherogenicity and obesity, validating our observations in part [40]. Although the link between lipid metabolism and variants *ARRB1* rs11236389 and *CYBA* rs12709102 identified here has not been previously reported, it is not totally unexpected since these genes or their paralogs seem to bear variants in relation to cardiovascular risk [1,40].

*TTN/CCDC141* rs10497525 is an intron variant of the large sarcomeric protein, titin; variations on this gene cause muscle disorders and cardiomyopathies [41]. *TTN*/*CCDC141* is highly expressed in the heart [42], suggesting its potential role in biochemical pathways and cardiovascular health, but it has not yet been discussed under the scope of dyslipidemia and atherogenic indexes. Our results may give rise to the biochemical connection between heart health, blood lipid levels, and genetics identified in adults under 53 years of age and free of cardiovascular disease.

The last variant associated with the AIP index was *KCND3* rs6703437, which codes for a potassium channel responsible for smooth muscle contraction and is associated with Brugada syndrome and cardiac conduction [43]. *KCND3* rs6703437 is 0.6 Kb apart and in partial LD with variant *KCND3* rs672757; this latter is directly associated with obesity in patients with asthma [44], hinting toward a potential role of heart disease under a lipid imbalance.

Overall and according to the recent literature, the variants associated with the AIP index contribute to the list of genetic variations potentially linked to cardiovascular health, heart function, lipid transport, and metabolism. Our observations confirm previous correlations between lipid levels and genes, *APOC3*, *TTN/CCDC141*, *KCND3*, *CYBA*, and *ARRB1,* and attest for the first time to a genetic relationship with AIP. We acknowledge that these associations do not mean causation, but that our results contribute to the collection of genetic variants that may partly explain atherogenicity in a quantitative manner.

### 4.2. Genetic Associations with Castelli Index 2, CI2=LDL-C/HDL-C

For CI2, we identified six variants on intergenic and non-coding loci, four of them on chromosome 12, with few or no reports on their clinical relevance. We identified *DIPK2B* rs4294309, an intron variant located on chromosome Xp11.3. *DIPK2B* codes for a protein kinase expressed in the endoplasmic reticulum, and it is involved in cellular protection and repair of cardiomyocytes through the PI3K-AKT-CDK7 pathway. Interestingly, this pathway involves the phosphorylation of lipids, which may support the association between *DIPK2B* rs4294309 and the LDL-C/HDL-C ratio [45].

Four variants on chromosome 12 were also associated with CI2, *LINC02451* rs6582413, *LINC02451* rs12817366, rs34115639, and rs10880344, with the two former on the Long Intergenic Non-Protein Coding RNA 2451 (*LINC02451*) and the two latter in intergenic regions and variants listed here. We may only speculate that regions on chromosome 12 could harbor gene variants related to CI2 and cardiovascular health or that this region may represent a polygenic cluster associated with atherogenesis.

Genetic variation on 10q21.3 rs7762658 and rs11251177, identified here, has been formerly associated with coronary artery disease in the GENOA study and in a pedigree of familial hypercholesterolemia [46]. This locus 10q.21.3 has been suggested to harbor genes with a role in subclinical coronary atherosclerosis [1], we identified this same locus as 10q21.3 rs1125117 with the highest statistical significance and size effect associated to CI2 (*p*-value 1.07 × 10^−7^). Lange et al. mentioned that this variant is enriched in families with hypertension, which is one of the future goals of the present cohort, i.e., the identification of markers predictive of hypertension and cardiovascular health. These observations highlight the relevance of cluster 10q21.3, in particular variant rs1125117 T > C. Extended investigations may model the quantitative impact of 10q21.3 rs1125117 on the CI2 ratio, which, after clinical validation, may be used to preemptively predict atherosclerosis risk.

Other variants associated with CI2 were observed on *KCND3*, *DIPK2B*, and *LIPC/ALDH1A2*, which have already been identified in lipoprotein and dyslipidemia studies. Here, *KCND3* rs6703437 was associated with both the AIP and CI2 indexes (*p*-value 2.06 × 10^−6^ and 1.7 × 10^−5^) suggesting a concomitant association of this variant with HDL-C. Current reports on *KCND3* indicate a link between genetic variation and cognitive impairment [47,48] suggesting a shared relationship between neurological health and lipids and hence the importance of monitoring genetic variation associated with lipid indexes as contributors to the assessment of overall health [26].

*LIPC/ALDH1A2* rs261342 was another variant associated with CI2. LIPC has a dual function as a triglyceride hydrolase and a ligand/bridging factor for receptor-mediated lipoprotein uptake. *LIPC/ALDH1A2* rs261342 is located on the 5′ promoter region of *LIPC* and has been strongly associated with HDL-C in women as part of a lipid-associated haplotype [49], and it is used to assess CVD risk in relation to lipids and apolipoproteins [50]. We found this variant associated with CI2, confirming its relationship to HDL-C and LDL-C, in both males and females. This coincides partly with previous observations by Feitosa et al. [49].

In summary, for C2I, we corroborate previous genetic associations with 10q21.3 rs11251177 and to loci clustered on chromosomes 12 and 6. We confirmed that variation on *LIPC* and *KCND3* may impact this atherogenic ratio and that *KCND3* may have a stronger relation to HDL-C and higher biomedical relevance, as we found it associated with both indexes. In summary, we confirmed previous reports on the relationship between genetic variation and the LDL-C/HLD-C ratio providing additional variants, such as rs34115639, rs10880344, or rs261342, that contribute to the genetic basis of atherogenesis.

The current literature shows only a few reports on the clinical impact of the loci here identified (Table 3) and its relation with lipids and lipid indexes; hence, to infer its functional consequence, we sought their in-silico impact in The Regulome database (visited on December 2022) [51]. For all variants listed (Table 3), including the most statistically significant, 10q21.3 rs11251177 and rs6582413, we found a neutral impact on clinical phenotypes. This leaves little room for interpretation and a void of information that ought to be addressed in future investigations.

This research faced some limitations, including the lack of clinical follow-up, a relatively small sample size, and a limited laboratory lipid analysis. Nevertheless, the experimental design, the genomic inclusion of 330,000 genetic variants, rigorous statistical strategies, and stringent model development provided enough statistical power to replicate previous genetic associations and novel variation that complements the current genetic knowledge of atherogenesis risk.

## 5. Conclusions

Here, we attest to the relationship between genetics and the atherogenic indexes AIP and CI2, which to the best of our knowledge has not been previously reported. We showed that the genetic variants listed here were associated to AIP and CI2 indexes overlapping previous reports of genetic associations to specific blood lipids. Our observations contribute to the list of loci linked to the assessment of atherogenic risk. The clinical utility of these SNVs as indicators of cardiovascular disease risk remains to be investigated, but their association with genetic variation highlights the genetic basis of atherogenesis and specific lipids [47]. Future studies should aim to integrate and validate a list of genetic markers consolidating knowledge on genetics and lipids that could be directed toward cardiovascular prevention.

## 6. Patents

No patents are filed or intended from the work reported in this manuscript.

## Figures and Tables

**Figure 1 genes-14-01214-f001:**
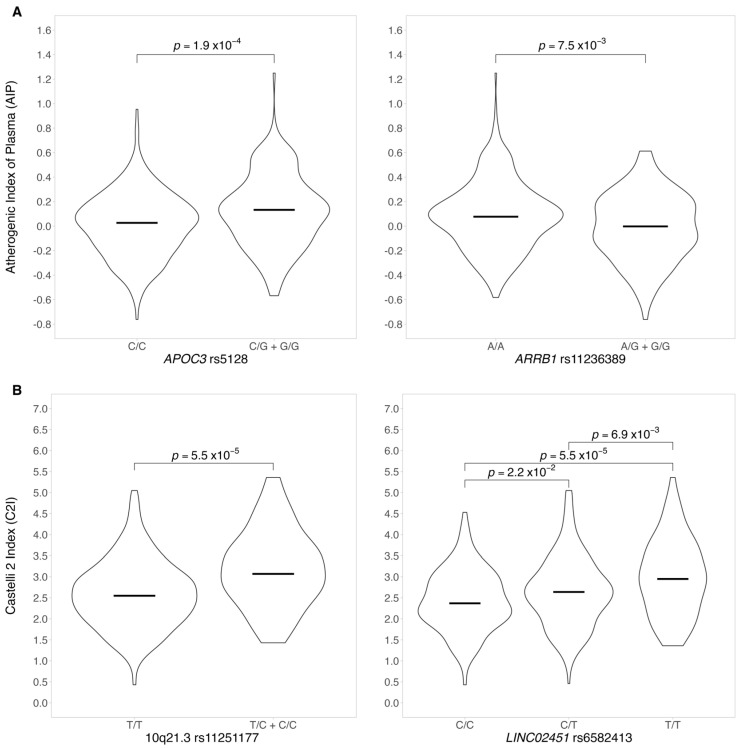
Impact of *APOC, ARRB1, LINC02451,* and *10q21.3* variants on AIP and CI2. (**A**). Association between AIP and *APOC3* rs5128 (**left**) and *ARRB1* rs11236389 (**right**); the horizontal line depicts the mean value. (**B**). Association between CI2 and 10q21.3 rs11251177 (**left**) and *LINC02451* rs65822413 (**right**); the horizontal line depicts the mean values. Comparisons shown here are statistically significant; *p*-values are displayed at the top.

**Table 1 genes-14-01214-t001:** Population characteristics.

	Males N = 170	Females N = 256	All N = 426
Age, y	38 (18–53)	40 (17–52)	39 (17–53)
Weight, kg	78.1 (51.2–125)	63.1 (41.7–119)	70 (41.7–125)
Height, m	1.70 (1.50–1.99)	1.57 (1.36–1.72)	1.61 (1.36–1.99)
BMI, kg/m^2^	26.8 (16.9–40.3)	26.2 (16.9–47.1)	26.4 (16.8–47.1)
Waist circumference, cm	94.0 (63.0–130)	85.0 (54.0–126)	89.0 (54.0–130)
Glucose, mg/dL	94.0 (72.0–166)	90.0 (74.0–241)	92.0 (72.0–241)
Uric acid, mg/dL	6.34 (1.82–10.0)	4.62 (2.30–7.58)	5.31 (1.82–10.0)
Creatinine, mg/dL	0.95 (0.62–1.40)	0.69 (0.44–1.19)	0.77 (0.44–1.40)
Cholesterol, mmol/dL	4.62 (2.96–8.30)	4.39 (2.16–7.06)	4.49 (2.16–8.30)
HDL-C, mmol/dL	1.10 (0.60–2.12)	1.22 (0.73–2.27)	1.16 (0.60–2.27)
LDL-C, mmol/dL	3.09 (0.98–6.87)	2.93 (0.54–5.38)	3.00 (0.54–6.87)
Triglycerides (TG), mmol/dL	1.48 (0.47–15.4)	1.22 (0.22–5.86)	1.32 (0.22–15.34)
Dyslipidemia, n (%)	119 (70%)	85 (72%)	304 (71%)
Castelli risk index 2 (CI2) ^1^	2.91 (1.25–5.36)	2.45 (0.43–4.92)	2.60 (0.43–5.36)
Atherogenic index of plasma (AIP) ^2^	0.48 (−0.18–1.61)	0.38 (−0.40–1.17)	0.42 (−0.40–1.61)

Values indicate the mean and range. ^1^ CI2 = LDL-C/HDL-C, ^2^ AIP = log (TG/HDL-C).

**Table 2 genes-14-01214-t002:** Proportion of lipid levels outside reference values.

	High TG > 1.9 mmol/L	High Cholesterol > 5 mmol/L	High LDL-C > 3.9 mmol/L	Low HDL-C < 1.04 mmol/L
^1^ Males %	30%, n = 51	35.3%, n = 60	16.5%, n = 28	38%, n = 65
^1^ Females %	16%, n = 41	22.3%, n = 59	10.9%, n = 28	26%, n = 67
*p*-value ^2^	8.74 × 10^−6^	2.32 × 10^−3^	1.80 × 10^−3^	2.59 × 10^−5^

^1^ n, sample size for each sex. ^2^
*p*-value for sex differences using an ANOVA test.

**Table 3 genes-14-01214-t003:** Variants associated with atherogenic indexes.

Gene	Chr	rs Identifier	Coefficient	*p*-Value
Variants associated with AIP				
*APOA1/APOC3*	11	rs5128, C > G	0.094	2.61 × 10^−6^
*CYBA*	16	rs12709102, T > C	0.078	3.91 × 10^−6^
*ARRIB1*	11	rs11236389, A > G	−0.102	6.63 × 10^−6^
*TTN/CCDC141*	2	rs10497528, A > C	0.089	8.29 × 10^−6^
*KCND3*	1	rs6703437	−0.177	0.90 × 10^−6^
*APOA1/APOC3*	11	rs5072, G > A	0.091	8.94 × 10^−6^
Variants associated with CI2				
*Intergenic*	10q21.3	rs11251177, T > C	0.606	1.07 × 10^−7^
*LINC02451*	12	rs6582413, T > C	0.259	5.19 × 10^−7^
*LINC02451*	12	rs12817366, C > T	0.254	1.88 × 10^−6^
*Intergenic*	12	rs34115639, C > T	0.244	7.06 × 10^−6^
*Intergenic*	12	rs10880344, T > C	−0.233	7.10 × 10^−6^
*Intergenic*	6	rs7762658, C > T	−0.247	2.03 × 10^−6^
*LIPC/ALDH1A2*	15	rs261342, C > G	0.227	1.10 × 10^−6^
*DIPK2B*	23	rs4294309, A > G	0.306	1.18 × 10^−5^
*KCND3*	1	rs6703437, G > A	−0.234	1.76 × 10^−5^

Lead SNVs generated by a generalized linear model (GLM) accounting for uric acid levels, weight, waist circumference, sex, and genetic ancestry. The raw output of the GLM analysis is in ST2.

## Data Availability

Data available on request from the corresponding author due to privacy restrictions and multiple institutions managing the future purpose of the cohort as a longitudinal study.

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
