# Peer review of "Genetic Determinants of Atherogenic Indexes"

_genes, 2023, doi:10.3390/genes14061214_

Round 1

Reviewer 1 Report

Dear Authors!
You do not indicate the alleles and/or genotypes of the studied rs. In the future, when citing, an ambiguous interpretation is possible, so I would like to see in Table 3 (or in a separate table) alleles and/or genotypes, not only "rs ids". Therefore, I think that your manuscript should be reconsidered after major revision.
I also have some small comments:
• In line 72 it is better to write (CI2=LDL-C/HDL-C) as in the abstract
• please check that all references in the text are in square brackets
• [24] is missing, is it not the same as ref. 26?
• also in 4.Discussion, reference 29 comes after 30, 31. Please double check the citation order.
Нope to see your article after the revision

Author Response

Dear Reviewer,

We appreciate the time and detailed observations to our manuscript. 

We addressed all your comments and suggestions. Below each observation, you will find our responses.

Dear Authors! You do not indicate the alleles and/or genotypes of the studied rs. In the future, when citing, an ambiguous interpretation is possible, so I would like to see in Table 3 (or in a separate table) alleles and/or genotypes, not only "rs ids". Therefore, I think that your manuscript should be reconsidered after major revision.

Thank you for your suggestion, Table 3 has been edited and now it includes alleles for each genotype.

I also have some small comments:
• In line 72 it is better to write (CI2=LDL-C/HDL-C) as in the abstract

We agree, we have made this change in page 2 line 72 and 96

  • please check that all references in the text are in square brackets
    • [24] is missing, is it not the same as ref. 26?
    • also in 4.Discussion, reference 29 comes after 30, 31. Please double check the citation order.
    Нope to see your article after the revision

We appreciate your observations regarding the references. We have revised references throughout the text and made the necessary changes.

Reviewer 2 Report

1. The justification for analysing genetic variants associated with these atherogenic indexes should be substantially changed throughout the text. First, these indexes have already proven their predictive value for ASCVD, have a low cost and can be easily and extensively applied in the general population. What is the clinical benefit of replacing or supplementing them with high-cost and limited availability genetic investigations? Second, the authors did not search at all for ASCVD, so speaking about the diagnostic value for ASCVD of the genetic variants they analysed is purely speculative.

2. Acronyms should be explained at their first mention in the text and not later or not at all – see CI2 (line 72 and not 97), LD (line 171), and GLM (line 177).

3. Information in lines 108-109 should find a better place in the Results section, not the Materials and Methods section. Similarly, parts of the phrase in lines 121-124 should belong to the Results and Discussion sections, and information in lines 167-168 and 172-173 – in the Discussion section.

4. The statistical procedures should be described in a distinct sub-section at the end of the Materials and Methods fragment.

5. The Conclusions are vague, redundant, allude to the same incorrect justifications mentioned at point 1, and therefore need a complete reformulation.  

6. What are the strengths and limitations of this study?

7. The whole manuscript would benefit from a thorough English language assessment. A series of syntactic errors (e.g., inappropriate use of the comma, inaccurate topic, some noun-verb disagreements, wrong verbal forms) undermines the background of an otherwise acceptable quality of the English language. Moreover, long-running phrases that bring together completely distinct notions and thus become confusing for the reader are one of the main limitations of the text. Some of these phrases can be found in lines 18-20, 20-24, 38-41, 62-65, 117-119, 207-209, 218-221, 229-234, 238-239, 244-249, 263-266, 275-281, 292-294, but an overall revision of the whole manuscript is required.

8. The following constructs and expressions are both formally and substantively unclear and should be revised:

“the characterisation of early molecular phenotypes atherosclerotic key events remain to be fully elucidated”

“It is possible that AIP and C2 ratios together with genetics could improve the clinical assessment of CVD risk and dyslipidemia.” How exactly is it possible?

“The identification of a quantitative relationship between genetics and CVD surrogates such as AIP and CI2 is of health transcendence due to the high mortality associated with cardiovascular disease underlied by atherogenesis.”

“adding genetic information to clinical CVD prevention may fine-tune the utility of lipid indexes for disease prediction” How exactly would this be useful?

“it might not necessarily be an unexpected observation since these genes, or their paralogs seem to bear variants in relation to cardiovascular risk”

“suggesting a potential lipid-gene-autism relationship that may possibly pinpoint to a genetic marker”

“We found this variant associated with CI2 confirming its relationship to HDL-C levels, in males and females, i.e., in a sex independent manner opposed to the sexual dimorphism listed in previous results.”

“For CI2, we corroborate previous associations between this atherogenic index and 10q21.3 rs11251177, and to loci clustered on chromosomes 12 and 6.” What did this corroboration result in?

“its association to genetic variation contributes to the not fully accounted genetic impact on lipids.”

“the genetic variants listed here are associated to these lipid indexes overlapping previous reports of genetic associations to specific blood lipids.”

Author Response

Dear reviewer,

Thank you for your comments and suggestions, they have helped us improve this manuscript.

Here you will find below each observation our response. These are also available as a pdf file attached. Thank you!

Reviewer 2

  1. The justification for analysing genetic variants associated with these atherogenic indexes should be substantially changed throughout the text. First, these indexes have already proven their predictive value for ASCVD, have a low cost and can be easily and extensively applied in the general population.

Yes, you have a point, these atherogenic indexes have been studied and show clinical utility. However, for researchers, their use and relation with genetics and other phenotypes is rather limited. This paper not only highlights the benefits of these indexes but its possible relationship with genetics. Since personalized medicine expects that individuals will have part or their whole genome sequenced in a preemptive manner it will be possible to predict or foresee certain metabolic conditions before they manifest or to deploy interventions to prevent the development of an associated phenotype.

To perform these predictions, it is important to depict, investigate and characterise genotype-phenotype associations, which may also reveal pathways not previously identified that could be of use in search of drug targets or preventive markers. Our results point to a) previously identified variants related to lipid levels offering some certainty of the validity of these observations and b) It lists variants not previously reported which could be related to other metabolic routes or to markers associated to the atherogenic phenotype. This was the case for this cohort. Initially we recruited only young healthy individuals with no diagnosis history to follow them for 15 years. However, 75% of these so-called healthy individuals showed dyslipidaemia. The markers here reported will need replication, validation, and a deeper investigation before one could become clinically useful. Nevertheless, this is a starting point that has not been reported before and that could contribute to the list of preemptive genetics.

We have modified the introduction and the beginning of the discussion to make this clear.

What is the clinical benefit of replacing or supplementing them with high-cost and limited availability genetic investigations?

Our goal is not to replace current clinical testing which, as you mentioned, is working and it is useful. The goal of this research is to identify genetic variants that may reveal additional biochemical pathways related to these indexes and to contribute to a genetic catalogue of variants that could partly explain the  atherogenic phenotype.  Moreover, our study may pinpoint to ethnic differences for basic research. In the future, this information would be of use for studies that aim to use molecular markers pre-emptively and predict a phenotype before the appearance of symptoms. Moreover, this research is biomedical and basic, it is hypothesis free, hence aiming to generate information and associations available for FUTURE investigations.

We have modified the introduction and the first paragraph of the discussion to emphasise the above as the aim of the study.

Second, the authors did not search at all for ASCVD, so speaking about the diagnostic value for ASCVD of the genetic variants they analysed is purely speculative.

We appreciate you brought this to our attention.

In the discussion, we have changed some text to diminish the impact of the adjectives used to suggest that these observations are readily useful. Also, we have revised all paragraphs of the discussion to make sure all inferences are stated as such i.e., as premises that will require investigation for a mechanistic characterization in addition to replication, validation, and definition of clinical utility.

It is worth mentioning that much research including this work has a basic science focus. It is oriented towards quantitative knowledge, and we are fully aware that not every observation has to reach the clinical lab.

A plethora of reports have found that the genetic determinants of lipids does contribute to the understanding of ASCVD but also helps explain biochemical pathways of individual lipids. For example, D. J. Raider et al highlighted the contribution of genetics (including genes here identified) to the comprehension of ASCVD and lipids (Clinical Implications of Lipid Genetics for Cardiovascular Disease Alanna Strong and Daniel J. Rader. https://www.ncbi.nlm.nih.gov/pmc/articles/PMC3155851/). It is not our goal to describe ASCVD comprehensively, but to increase the genetic information potentially associated with lipid indexes used to assess and diagnose ASCVD.

Our contribution is under the realm of genetic-phenotype associations which is mathematical and does not mean causation. Our attempts are to increase knowledge for a highly prevalent and costly health problem.

  1. Acronyms should be explained at their first mention in the text and not later or not at all – see CI2 (line 72 and not 97), LD (line 171), and GLM (line 177).

Thank you for your observations. Changes have been made.

  1. Information in lines 108-109 should find a better place in the Results section, not the Materials and Methods section. Similarly, parts of the phrase in lines 121-124 should belong to the Results and Discussion sections, and information in lines 167-168 and 172-173 – in the Discussion section.

Thank you for your observations. Changes have been made including the relocation of text from materials and methods to Results.

  1. The statistical procedures should be described in a distinct sub-section at the end of the Materials and Methods fragment.

Statistical analyses were detailed in section 2.3

  1. The Conclusions are vague, redundant, allude to the same incorrect justifications mentioned at point 1, and therefore need a complete reformulation.  

Based on the limitations of the study we have rephrased the rationale and conclusions. We are positive that this new version better explains the purpose of this research which by no means is to replace current metrics, but to expand knowledge from basic research.

  1. What are the strengths and limitations of this study?

We included in the text a summary with the following strengths and limitations of the study:

The size of this study group N=445 provided enough statistical power for the identification of genetic variants that may associate with blood lipids and atherogenic indexes, an association that has not been clearly defined and limited information is available for populations from Latin America. It also contributes to the collection of genetic information associated with markers of prevalent diseases. This research also extends current genetic-phenotype knowledge for an admixed population of Mexican descent where strategies and interventions are urgently needed to improve the high prevalence of CVD.

This research faced several limitations including the lack of follow up on the study group, and the limited number of lab tests performed that may decrease the probability of identifying genotype-phenotype associations. An additional limitation may be the scope of the project which was directed towards the identification of genetic variants and  its relation to blood lipid indexes. Future endeavours may seek to expand these analyses and include genetic imputation, to complement current associations with biochemical and transcription parameters.

Comments on the Quality of English Language

  1. The whole manuscript would benefit from a thorough English language assessment. A series of syntactic errors (e.g., inappropriate use of the comma, inaccurate topic, some noun-verb disagreements, wrong verbal forms) undermines the background of an otherwise acceptable quality of the English language. Moreover, long-running phrases that bring together completely distinct notions and thus become confusing for the reader are one of the main limitations of the text. Some of these phrases can be found in lines 18-20, 20-24, 38-41, 62-65, 117-119, 207-209, 218-221, 229-234, 238-239, 244-249, 263-266, 275-281, 292-294, but an overall revision of the whole manuscript is required.

  1. The following constructs and expressions are both formally and substantively unclear and should be revised:

Thank you for your observations. We have adjusted most of these phrases to simplify them and offer a more direct context.

“the characterisation of early molecular phenotypes atherosclerotic key events remain to be fully elucidated”

Modified

“It is possible that AIP and C2 ratios together with genetics could improve the clinical assessment of CVD risk and dyslipidemia.” How exactly is it possible?

You are right, we have modified the text it is not directly possible

“The identification of a quantitative relationship between genetics and CVD surrogates such as AIP and CI2 is of health transcendence due to the high mortality associated with cardiovascular disease underlined by atherogenesis.”

“adding genetic information to clinical CVD prevention may fine-tune the utility of lipid indexes for disease prediction” How exactly would this be useful?

You are right, we have modified the text in the scope of basic research and future preemptive medicine

“it might not necessarily be an unexpected observation since these genes, or their paralogs seem to bear variants in relation to cardiovascular risk”

“suggesting a potential lipid-gene-autism relationship that may possibly pinpoint to a genetic marker”

“We found this variant associated with CI2 confirming its relationship to HDL-C levels, in males and females, i.e., in a sex independent manner opposed to the sexual dimorphism listed in previous results.”

“For CI2, we corroborate previous associations between this atherogenic index and 10q21.3 rs11251177, and to loci clustered on chromosomes 12 and 6.” What did this corroboration result in?

“its association to genetic variation contributes to the not fully accounted genetic impact on lipids.”

“the genetic variants listed here are associated to these lipid indexes overlapping previous reports of genetic associations to specific blood lipids.”

Round 2

Reviewer 1 Report

The authors took into account the main remarks. At the moment I have no complaints about the content of the text, except for a few remarks:

• Males showed higher total TG (49%), LDL-C 32%,
> omitted parentheses

• 4.2 Genetic associations with the Castelli index 2
> note that the paragraph must be placed on a new line

And in the list of references, for ref. 8 the year is missed, and for ref. 47 all the data (not only year) is missed.

There are also some comments on the quality of english language (see the relevant section)

I think that now manuscript needs only text editing and next round of review is not required.

The placement of punctuation marks is questionable, so should in the sentences "including, genetics, the environment, low-density lipoprotein cholesterol" and "and CVD including, APOC3/APOA1", "several lipid traits including, HDL-C and TG" be a comma after "including"?

And somewhere plurals are missing:
• Atherosclerosis is a complex and multifactorial diseases
> disease
• Genetic variation on 10q21.3
> variations

Author Response

Thank you for your comments and suggestions and for bringing to our attention, the spelling and punctuation mistakes.

We have revised and corrected, the reviewer’s remarks, made spelling and punctuation modifications accordingly.

References 7 and 8 were checked and reinserted with a full reference.

We also made changes in response to the comments on the English language section.

  • Males showed higher total TG (49%), LDL-C 32%,
    > omitted parentheses
  • 4.2 Genetic associations with the Castelli index 2
    > note that the paragraph must be placed on a new line

    And in the list of references, for ref. 8 the year is missed, and for ref. 47 all the data (not only year) is missed.

    There are also some comments on the quality of English language (see the relevant section)

    I think that now manuscript needs only text editing and next round of review is not required.

We have also addressed the comments on the quality of English language.

Attached we share a highlighted-track changes file to attest for these modifications.

Reviewer 2 Report

1. Please explain the CI2 acronym at its first mention in the text and not later (line 72 and not 97). The LD acronym should be introduced when first mentioning the linkage disequilibrium (line 129 and not 223).

2. Some information about statistical procedures is still not included in the 2.3 subsection.

The manuscript still requires a thorough English language assessment involving the contribution of a native English speaker to advise the authors about the proper break of the long-running, unintelligible phrases that bring together completely distinct notions and thus become confusing for the reader. Unfortunately, such fragments are frequently seen throughout the manuscript and do not seem to have been appropriately approached by the authors. Please see my comments in the previous review form. Moreover, some syntactic errors I have previously warned the authors about (e.g., inappropriate comma use, inaccurate topic, some noun-verb disagreements and wrong verbal forms) were not yet corrected.

Author Response

Reviewer 2

Thank you for your comments and suggestions. They have contributed to substantially imrpove this manuscript. We have addressed each comment, below you’ll find a description in unerlined font of each of the changes suggested for this and the previous review round. Also, we share a filed attached in track-changes mode showing all the modifications to the manuscript.

  1. Please explain the CI2 acronym at its first mention in the text and not later (line 72 and not 97). The LD acronym should be introduced when first mentioning the linkage disequilibrium (line 129 and not 223).

Yes, thank you we thought we had addressed this abbreviation in the previous review. Now it should be clear.

  1. Some information about statistical procedures is still not included in the 2.3 subsection. The highlighted version of the manuscript shows that all the methods section lines 90 to 161 have been modified in an attempt to be more descriptive of the statistical procedures. Moreover, these references where authors perform similar statistical analyses were added:
  2. Lee J, Lee S, Min JY, Min KB. Association between Serum Lipid Parameters and Cognitive Performance in Older Adults. J Clin Med. 2021;10(22):5405. doi:10.3390/JCM10225405
  3. Uffelmann E, Huang QQ, Munung NS, et al. Genome-wide association studies. Nat Rev Methods Prim 2021 11. 2021;1(1):1-21. doi:10.1038/s43586-021-00056-9
  4. Jurado-Camacho PA, Cid-Soto MA, Barajas-Olmos F, et al. Exome Sequencing Data Analysis and a Case-Control Study in Mexican Population Reveals Lipid Trait Associations of New and Known Genetic Variants in Dyslipidemia-Associated Loci. Front Genet. 2022;13:807381. doi:10.3389/FGENE.2022.807381/FULL

Comments on the Quality of English Language

The manuscript still requires a thorough English language assessment involving the contribution of a native English speaker to advise the authors about the proper break of the long-running, unintelligible phrases that bring together completely distinct notions and thus become confusing for the reader. Unfortunately, such fragments are frequently seen throughout the manuscript and do not seem to have been appropriately approached by the authors.

-- As a team we went through the manuscript a couple more times. Punctuation has been revised, commas placed correctly, long sentences broken down to smaller ideas. We are sharing a highlighted version with revisions, so the reviewer realizes that we have paid attention to all the comments which are indeed very valuable.

--Our first revision did include a version revised by an English native speaker. We have asked him again to go through each of your suggestions and several more changes were made. We list each of them according to the list of the first review that you sent us.

REVIEWER 2 FIRST ROUND REMARKS

long-running phrases that bring together completely distinct notions and thus become confusing for the reader are one of the main limitations of the text. Some of these phrases can be found in lines 18-20, 20-24, 38-41, 62-65, 117-119, 207-209, 218-221, 229-234, 238-239, 244-249, 263-266, 275-281, 292-294, but an overall revision of the whole manuscript is required.

  1. The following constructs and expressions are both formally and substantively unclear and should be revised:

“the characterisation of early molecular phenotypes atherosclerotic key events remain to be fully elucidated”

This phrase has been removed and the idea rewritten in appropriate context.

“It is possible that AIP and C2 ratios together with genetics could improve the clinical assessment of CVD risk and dyslipidemia.” How exactly is it possible?

This phrase now reads: It is possible that genetic variants associated to AIP and C2 ratios could hint towards biochemical paths linked to the development of CVD risk and dyslipidemia. Lines 74-76

“The identification of a quantitative relationship between genetics and CVD surrogates such as AIP and CI2 is of health transcendence due to the high mortality associated with cardiovascular disease underlied by atherogenesis.”

This phrase has been modified and now reads: The identification of a quantitative relationship between genetics and CVD surrogates such as AIP and CI2 is of health transcendence due to the prominent role of atherogenesis in cardiovascular disease. (line 262)

“adding genetic information to clinical CVD prevention may fine-tune the utility of lipid indexes for disease prediction” How exactly would this be useful?

This sentence has been modified now reads: Evidence from other fields suggests that adding genetic information to clinical interpretation may support the utility of genetic variants as predictors of atherogenesis 19 Line 284

“it might not necessarily be an unexpected observation since these genes, or their paralogs seem to bear variants in relation to cardiovascular risk”

This sentence has been modified to: “it is not totally unexpected since these genes, or their paralogs seem to bear variants in relation to cardiovascular risk 1,41. “ Line 312

“suggesting a potential lipid-gene-autism relationship that may possibly pinpoint to a genetic marker”

The above phrase has been removed, deeper research done about DIPK2B and this phrase now reads: DIPK2B codes for a protein kinase expressed in the endoplasmic reticulum, it is involved in cardiac health through cellular protection and repair mechanisms. These protective mechanisms seem to be mediated through the PI3K-AKT-CDK7 pathway, responsible for the phosphorylation of different lipids  46. Line 345

We found this variant associated with CI2 confirming its relationship to HDL-C levels, in males and females, i.e., in a sex independent manner opposed to the sexual dimorphism listed in previous results.”

This phrase has been modified and now it reads: We found this variant associated with CI2 confirming its relationship to HDL-C and LDL-C, in both males and females. It coincides partly with previous observations by Feitosa et al. 52

“For CI2, we corroborate previous associations between this atherogenic index and 10q21.3 rs11251177, and to loci clustered on chromosomes 12 and 6.” What did this corroboration result in?

This phrase has been modified significnalty and it reads: In summary for C2I, we corroborate previous genetic associations with 10q21.3 rs11251177, and to loci clustered on chromosomes 12 and 6. We confirmed that variation on LIPC and KCND3 may impact this atherogenic ratio, and that KCND3 may have a stronger relation to HDL-C and higher biomedical relevance as we found it associated with both indexes.  In summary, we confirmed previous reports on the relationship between genetic variation and the LDL-C/HLD-C ratio providing additional variants such as rs34115639, rs10880344, or rs261342 that contribute to the genetic basis of atherogenesis

“its association to genetic variation contributes to the not fully accounted genetic impact on lipids.” “the genetic variants listed here are associated to these lipid indexes overlapping previous reports of genetic associations to specific blood lipids.”

The conclusions section has been significantly modified and the new text is on lines 488-496
